

# Automated snow avalanche release area delineation - validation of existing algorithms and proposition of a new object-based approach for large scale hazard indication mapping

Yves Bühler[1], Daniel von Rickenbach[1/2], Andreas Stoffel[1], Stefan Margreth[1], Lukas Stoffel[1], Marc Christen[1]

[1]WSL Institute for Snow and Avalanche Research SLF, Davos Dorf, 7260, Switzerland
[2]Department of Geography, University of Zürich, Zürich, 8057, Switzerland

*Correspondence to*: Yves Bühler (buehler@slf.ch)

**Abstract.** Snow avalanche hazard is threatening people and infrastructure in all alpine regions with seasonal or permanent snow cover around the globe. Coping with this hazard is a big challenge and during the past centuries, different strategies were developed. Today, in Switzerland, experienced avalanche engineers produce hazard maps with a very high reliability based on avalanche cadastre information, terrain analysis, climatological datasets and numerical modelling of the flow dynamics for selected avalanche tracks that might affect settlements. However, for regions outside the considered settlement areas such area-wide hazard maps are not available mainly because of the too high cost, in Switzerland and in most mountain regions around the world. Therefore, hazard indication maps, even though they are less reliable and less detailed, are often the only spatial planning tool available. To produce meaningful and cost-effective avalanche hazard indication maps over large regions (regional to national scale), automated release area delineation has to be combined with volume estimations and state-of-the-art numerical avalanche simulations.

In this paper we validate existing potential release area (PRA) delineation algorithms, published in peer-reviewed journals, that are based on digital terrain models and their derivatives such as slope angle, aspect, roughness and curvature. For validation, we apply avalanche cadastre data from three different ski resorts in the vicinity of Davos, Switzerland, where experienced ski-patrol staff mapped most avalanches in detail since many years. After calculating the best fit input parameters for every tested algorithm, we compare their performance based on the reference datasets. Because all tested algorithms do not provide meaningful delineation between individual potential release areas (PRA), we propose a new algorithm based on object-based image analysis (OBIA). In combination with an automatic procedure to estimate the average release depth (d0), defining the avalanche release volume, this algorithm enables the numerical simulation of thousands of avalanches over large regions applying the well-established avalanche dynamics model RAMMS. We demonstrate this for the region of Davos for two hazard scenarios, *frequent* (10 – 30 years return period) and *extreme* (100 – 300 years return period). This approach opens the door for large scale avalanche hazard indication mapping in all regions where high quality and resolution digital terrain models and snow data are available.





## 1    1 Introduction

Snow avalanches are a severe threat in alpine regions around the world endangering people, buildings and traffic infrastructure. In Switzerland an average of 25 people die per year in avalanches, the vast majority during winter sport activities (Techel et al., 2015) and avalanches often cause infrastructure damage. In winter 1999 the total damage was more than 500 million Euros

(SLF, 2000). Switzerland has long-term experience coping with avalanche hazard. These range from spatial planning measures such as avoid to build where there is an avalanche hazard, usually achieved by trial and error over centuries up to constructional measures such as the splitting wedge at the church of Davos Frauenkirch, built in 1603 after the previous church was destroyed by a large avalanche. One of the most important measures is the generation of hazard maps. Such maps, the first one was already released 1953 in the Bernese Alps, are based today on avalanche cadastre information, climatic information on extreme

snow fall events, terrain analysis and numerical simulations of the avalanche dynamics. All this information is combined by experienced experts into scenarios. In Switzerland, hazard maps are based on 30-, 100- and 300-year scenarios (Figure 11a). Hazard maps show the hazard degree based on the frequency and intensity of avalanches. The elaboration of hazard maps is very demanding with respect to time and expertise. Furthermore it can only be applied to single avalanche tracks that are in particular endangering critical infrastructure (Gruber and Margreth, 2001). But these maps are the backbone of the Swiss

avalanche mitigation strategy and are legally binding where new infrastructure can be built and where not.

Hazard indication maps on the other hand are less detailed and accurate than hazard maps but can give a spatial continuous overview on avalanche hazards based on numerical simulations over large regions. Hazard indication maps are based on an extreme scenario and do not show different hazard degrees. This is in particular useful for regions with sparse avalanche

cadastre information, which is the case for most alpine regions around the world. High spatial resolution digital terrain models (DTMs) generated from remote sensing instruments get more and more available for mountain regions (Bühler et al., 2012;Fonstad et al., 2013;ASPRS, 2001) and open the door for meaningful numerical avalanche simulations (Bühler et al., 2011) over large areas, e.g. entire valleys, states or even countries.

To perform dynamic avalanche simulations with state-of-the-art software such as RAMMS (Christen et al., 2010) or SAMOS (Sampl and Zwinger, 2004) an accurate identification of the release areas and the release volumes is mandatory. Prerequisites for triggering a snow slab avalanche can be summarized in three categories: (1) terrain, (2) snow cover specific and (3) meteorological factors (Schweizer et al., 2003). In the past, different algorithms have been developed to identify potential snow avalanche release areas (PRA) mainly based on terrain specific parameters (1). So far, these algorithms to identify PRA

have never been compared and tested against a large reference dataset of observed and carefully mapped avalanche release areas. Therefore, the first aim of this study is to conduct a comparison of the existing algorithms based on avalanche cadastre information from Davos, Switzerland. Based on the findings gained from this comparison, we develop a new algorithm for automated PRA delineation and validate it in three selected test sites around Davos, Switzerland where we have an excellent





avalanche cadastre. We constrain the reference dataset to slab avalanches and exclude loose snow avalanches, which can start at point locations also in very steep terrain but do not mobilize masses that could be dangerous for people and infrastructure in the Alps. However, such avalanches can become more dangerous in other regions such as the Chilean Andes (Vera Valero et al., 2016).

## 1.1 Existing algorithms

In the past, DTM-based identification of PRA for different types of mass movements have been developed, in particular for shallow and deep-seated landslides (Carrara, 1983;Carrara and Guzetti, 1995;Singh et al., 2005;Gruber et al., 2009;Pradhan and Buchroithner, 2012) as well as for debris flows and rock falls (Singh et al., 2005;Michoud et al., 2012). The most important DTM-derived parameter for landslides and rock falls is the slope angle, which strongly determines the distribution of unstable areas (Carrara and Guzetti, 1995). In addition, also aspect and curvature are considered (Singh et al., 2005;Pradhan and Buchroithner, 2012).

Similar attempts have been made to automatically delineate snow avalanche release areas. Voellmy (1955) already stated that the terrain parameter slope angle plays a decisive role to identify PRA. The first automated approaches to identify PRA, considering different terrain parameters, began with the availability of DTMs with quite coarse resolution in the range of 25 to 30 meter (Maggioni et al., 2002;Maggioni, 2005;Maggioni and Gruber, 2003). DTMs with higher spatial resolution (1 to 10 m) enable the calculation of DTM-derivatives such as ruggedness or curvature, which are of major importance for avalanche release (van Herwijnen and Heierli, 2009;McClung, 2001;Schweizer et al., 2003;Vontobel, 2011). Table 1 gives an overview on PRA delineation algorithms, published in peer-reviewed web of science journals, and the terrain derivatives they apply.

**Table 1: Overview on the published avalanche release area delineation algorithms including the applied parameters**

| Publication (chronologic) | Slope | Plan curvature | Profile Curvature | Roughness | Aspect | Elevation | Distance to ridge | Wind Shelter | Topographic Wetness Index | Vegetation |
|---|---|---|---|---|---|---|---|---|---|---|
| Voellmy (1955) | x | | | | | | | | | |
| Maggioni et al. (2002) | x | x | | | x | | x | | | |
| Ghinoi & Chung (2005) | x | | | (x) | x | x | x | | | (x) |
| Barbolini et al. (2011) | x | x | | | | | | | | x |
| Andres & Cìa (2012) | x | x | | | | x | | | | x |
| Pistocchi & Notarnicola (2013) | x | x | x | | x | | x | | x | x |
| Bühler et al. (2013) | x | x | | x | | | | | | |
| Chueca Cìa et al. (2014) | x | x | | | | x | | | | x |
| Veitinger et al. (2016) | x | | | x | | | | x | | x |

All listed algorithms apply the parameter slope, most apply plan curvature but only two of them also include the parameter roughness. For this study, the following three existing algorithms are compared and validated: one simple algorithm





considering just the parameter slope after Voellmy (1955) as benchmark and the two recent algorithms taking the terrain roughness into account (Bühler et al., 2013;Veitinger et al., 2016). The algorithm of Maggioni et al. (2002) was also tested but only produces meaningful results with DEM resolutions in the order of 25 m. Furthermore, this algorithm was written in AML script language and cannot be run on current software anymore. The other algorithms by Ghinoi and Chung (2005);Barbolini

et al. (2011);Andres and Chueca Cia (2012);Pistocchi and Notarnicola (2013);Chueca Cía et al. (2014) have been developed by other research groups and were not available for comparison.

## 2 Validation of selected existing algorithms

### 2.1 Reference dataset

For a meaningful comparison and validation of the three selected algorithms, a good reference dataset is mandatory. Explicit

and accurate reference data on avalanche release areas are very scarce because the release areas are remote, mostly in poorly accessible terrain. Even though some approaches exist to automatically map snow avalanches from high spatial resolution remote sensing data (Bühler et al., 2009;Lato et al., 2012;Eckerstorfer et al., 2016;Korzeniowska et al., 2017), such datasets are only available for isolated time periods. Recent advances using freely available Sentinel-1 radar data succeed to map a part of the avalanche deposits but do not produce reliable information on release areas (Eckerstorfer et al., 2017). Therefore, the

best available information on avalanche release areas still comes from manual mapping in the field.

For the region of Davos, Switzerland, the SLF maintains an event cadastre whereby the avalanche contours and the release areas are mapped by experienced staff. This event cadastre is considered to be the best reference data set available today. The reference dataset contains 5785 mapped release areas from the year 1970 until 2016. The avalanches of the extreme winter

1999 are as well included in the data set. Out of this reference dataset, three test sites were deliberately chosen: Parsenn, Jakobshorn and Rinerhorn. These test sites are located in the three largest ski resorts of Davos. Consequently, these areas are surveyed more or less constantly during winter operations and observed avalanche events, naturally and artificially triggered, are mapped by the ski patrol and included into the SLF database. We limit the test regions to the areas which are well observed from the ski resorts and exclude terrain sections where observations of avalanche are difficult. Thus, it can be assumed that,

as far as possible, all potential release areas have already been mapped at least once. However, a check of the reference data set with the local ski patrol heads showed, that certain release areas known to them are still not included. We included these observations also into the reference dataset at the three test sites to achieve a result as complete as possible.

The test site Parsenn (Figure 1) is the largest with an area of 7.3 km$^2$ ranging from 2200 m a.s.l. up to 2830 m a.s.l. with a

mean elevation of 2460 m a.s.l.. The mean slope angle is 25°. In this region we count 1382 individual manually mapped PRA which do partly overlap. In the southern part, a small area is spared out because it is not well visible from the ski area and therefore not well enough documented. The test site Jakobshorn (Figure 1) has an area of 2 km$^2$ and ranges from 2310 m a.s.l.





up to 2680 m a.s.l. with a mean elevation of 2450 m a.s.l.. The mean slope angle is 28° which is considerably steeper than Parsenn but the terrain does lack steep rock faces. We count 309 individual PRA at Jakobshorn. The test region Rinerhorn (Figure 1) ranges from 2200 m a.s.l. up to 2910 m a.s.l. with a mean elevation of 2440 m a.s.l. and a mean slope angle of 29°. It is the steepest test region with the largest amount of rough rock faces and counts 438 individual PRA. In total we have 2129

individual PRA in all three test sites.

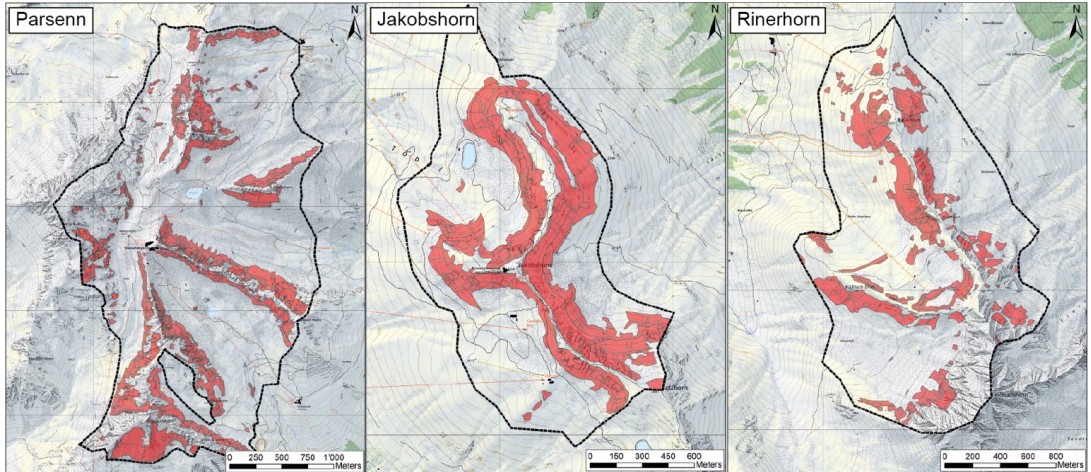

**Figure 1: Reference datasets including the manually mapped avalanche release areas (red polygons) in the three test regions Parsenn, Jakobshorn and Rinerhorn (pixmaps©2018 swisstopo (5704 000 000), reproduced by permission of swisstopo (JA100118).**

## 2.2 Validation Methods

The three selected algorithms subdivide the area within a certain perimeter into the two classes: *potential release area* (PRA) and *no potential release area* (NoPRA). The model output of these algorithms can be seen as thematic maps with the two discrete classes PRA and NoPRA. Hence, for validation an error matrix (Figure 2) is calculated as it is known for the accuracy assessment of thematic mapping in remote sensing (Congalton and Green, 1999). To produce an error matrix, a reference dataset is needed. Each value in the error matrix represents the total intersectional area of a specific reference class and

algorithm class. For most of the state-of-the-art avalanche dynamic simulation software, potential release areas are required as discrete vector objects to be able to perform dynamic avalanche simulations. Therefore, the algorithms also output the release areas as vector objects. Consequently, the area values calculated for the error-matrix are square meters and not number of pixels, which would be rather common for the error matrix (Stehman and Wickham, 2011). Based on this error matrix, different accuracy measures can be calculated (Allouche et al., 2006):

*Probability of Detection* $(POD) = \frac{a}{a+c}$ (1)

   *Probability of False Detection* $(POFD) = \frac{b}{b+d}$ (2)





$$Peirce\ Skill\ Score\ (PSS) = POD - POFD = \frac{a}{a+c} - \frac{b}{b+d} \tag{3}$$

$$Overall\ Accuracy\ (OA) = \frac{a+d}{n} \tag{4}$$

$$Heidke\ Skill\ Score\ (HSS) = \frac{OA - CA}{1 - CA} \quad, \quad CA = \frac{(a+b \times a+c) + (c+d \times b+d)}{n^2} \tag{5}$$

| | reference | | |
|---|---|---|---|
| | **PRA** | **NoPRA** | **Σ** |
| **PRA** | $a_1 + a_2 = a$ | $b_1 + b_2 = b$ | **a+b** |
| **NoPRA** | **c** | **d** | **c+d** |
| **Σ** | **a+c** | **b+d** | **n** |

*(algorithm — row label on left side)*

**Figure 2: Schema of the applied validation measures after Congalton and Green (1999)**

The Peirce Skill Score (PSS) is applied as accuracy measure to find the optimal values for the input parameters. The PSS is also known as True Skill Statistics (TSS) or Hanssen and Kuipers discriminant named after Hanssen and Kuipers (1965). In
this paper, the PSS is scaled by a factor 100 to enhance the interpretability and thus ranges from -100 to 100. A PSS of 100 indicates perfect agreement, whereas values of zero or less indicate a performance not better or worse than a random classifier (Allouche et al., 2006). As aforementioned, the purpose of possible release area delineation in this paper is the automated hazard indication mapping. Hazard indication maps should comprise all potential spatial occurrences of a certain hazards process (in this case snow avalanches). Therefore, the algorithm should correctly classify PRA compared to the reference
dataset. But at the same time, NoPRA area should also be correctly classified. The first criterium is measured by the Probability of Detection (POD) and the second by the Probability of False Detection (POFD). The better the algorithm the higher POD (range: 0-100) and the lower the POFD (range: 0-100). These two measures are combined in the Peirce Skill Score (PSS). In contrary to the HSS, the PSS is independent to prevalence (Allouche et al., 2006). Prevalence stands for different proportion of the two classes (in this case PRA and NoPRA) (Beguería, 2006). Therefore the PSS is widely applied for the validation and
evaluation of predictive models in hazard assessment and risk management (Beguería, 2006). Examples for this are the validation of landslide susceptibility models (Leonarduzzi et al., 2017;Gariano et al., 2015;Frattini et al., 2010) or avalanche hazard (Purves et al., 2003). As a second overall accuracy measure we apply the Heidke Skill Score HSS (Heidke, 1926), also referred to as Kappa.





## 2.3 Input parameter setting

The algorithm created by Veitinger et al. (2016) requires parametrization for *snow depth*, *wind direction* and w*ind direction variability*. We specify no dominant wind direction as this cannot be clearly identified as the region faces different wind regimes ranging from northwest to south (Schüepp and Urfer, 1962). Furthermore, in most regions a broad hazard indication

mapping scenario cannot be reliably connected to a specific wind regime. The roughness is calculated depending on the mean snow depth. Afterwards, the algorithm applies an individual fuzzy membership function to roughness and slope. Based on this multi-scale, fuzzy logic approach, the algorithms output indicates the avalanche release probability in a continuous range from 0 (not probable) to 1 (highly probable). The other algorithms apply Boolean Classifiers and thus these outputs exhibit the two discrete classes 0 (NoPRA) and 1 (PRA). Additionally, they have the option to eliminate PRA with an area smaller than a

certain threshold. Therefore, the algorithm of Veitinger et al. (2016) was extended by the option to set a probability threshold for discrete classification and the option to define a minimum release area. The algorithm by Bühler et al. (2013) requires a value for the following input parameters: *minimum slope angle, maximum slope angle, cell size for the moving window to calculate the roughness, maximum roughness, maximum curvature, minimum release area*. To run the algorithms, we apply the swissalti3D digital elevation model from swisstopo with a spatial resolution of 5 m (swisstopo, 2018).

Each of the selected existing algorithms requires certain values as input parameters. Depending on the value set for the input parameters, the model output varies considerably. In order to be able to compare the algorithms with each other, the following approach is chosen: The goal is to achieve the best possible performance of each algorithm and to compare it with the other algorithms. Therefore, the aim is to find the optimal values for each input parameter. To do so, for an algorithm the values of

a parameter are changed systematically at a time (e.g. 20°, 21°, 22° up to 40° for the lower slope angle threshold) while keeping all other input parameters of this algorithm fixed (e.g. 60° for the upper slope angle threshold). After the iteration over this parameter (lower slope angle threshold), another input parameter (e.g. upper slope angle threshold) is changed systematically while the lower slope angle threshold is fixed again to the "default value" (e.g. 30°). With this approach we identify the optimal parameter settings for the tested algorithms based on the same reference dataset described in section 2.1. This identification of

the optimal parameter settings (Figure 3) enables an objective comparison of the performance of the different algorithms (section 2.4) applying the quality measures described in section 2.2 even though they apply different input parameters.



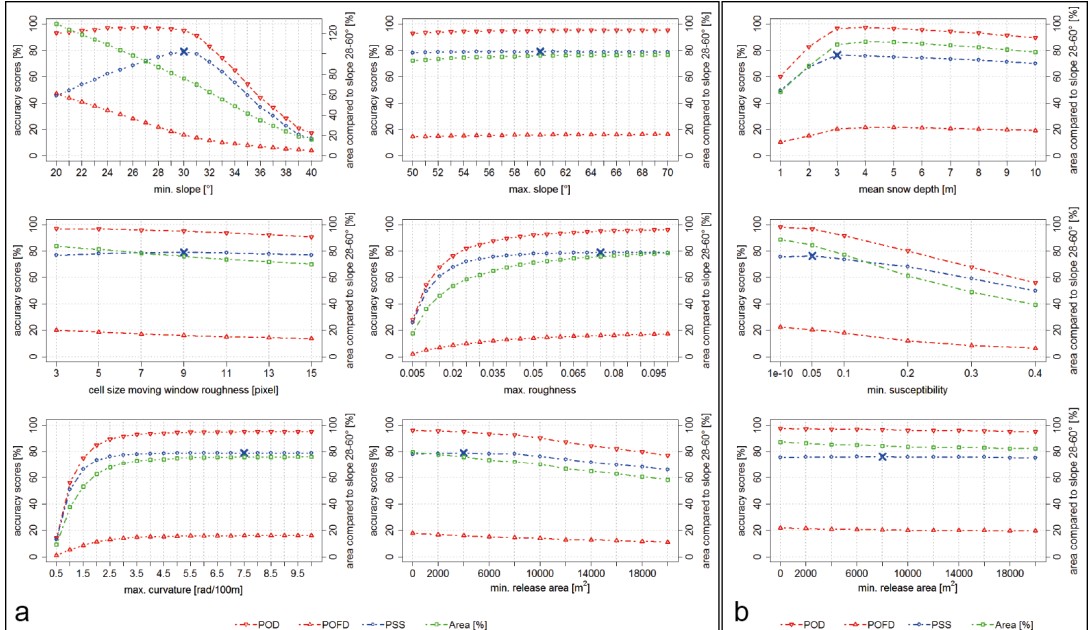

**Figure 3: Identification of the optimal input parameter settings for the selected algorithms a) Bühler et al. (2013) and b) Veitinger et al. (2016). The blue crosses indicate the derived optimal parameter values. The accuracy measures POD, POFD and PSS are depicted in the left axis. The percentage of identified area compared to the benchmark after (Voellmy, 1955) is corresponding to the right axis.**

Some parameters such as the maximum slope angle or minimum release area are not sensitive and do only slightly change with parameter variation. Other parameters such as the minimum slope angle or the minimum susceptibility are much more sensitive and change the output considerably. The minimum slope angle in Figure 3a reaches percentage values of more than 100% (right axis) because we test minimum slope angles smaller than 28°.

## 2.4 Validation Results

We compare the performance of the selected algorithms with their specific optimal parameter settings (Table 2). To quantify the optimization effect of the algorithms we also list the percentage of PRA area delimited by the respective algorithm to the PRA area identified in the slope only approach (Voellmy, 1955) which corresponds to all area between 28° and 60° slope angle. The slope only approach has a very high probability of detection (POD) but also a very high probability of false detection (POFD) leading to a relatively low PSS and HSS value, because it basically delineates all area where avalanches can release. Even though the POD of the Veitinger et al. (2016) algorithm is 1.95 % lower, the PSS is 6.44% higher because the POFD is 8.39% higher, which is a substantial improvement. In comparison, the Bühler et al. (2013) algorithm reaches a slightly lower


POD of 95.06% but the highest POFD with 16.06%. This leads to the highest overall quality scores of 79.01% PSS and 66.99% HSS.

**Table 2: Performance of the tested algorithms compared to the manually mapped avalanche release areas in all three test sites Parsenn, Jakobshorn and Rinerhorn (Figure 1).**

|  | POD [%] | POFD [%] | PSS [%] | HSS [%] | Area compared to slope 28-60° [%] |
|---|---|---|---|---|---|
| **Vollmey (1955), slope only** | 98.69 | 28.89 | 69.80 | 51.72 | 100.00 |
| **Bühler et al. (2013)** | 95.06 | 16.06 | 79.01 | 66.99 | 75.91 |
| **Veitinger et al. (2016)** | 96.74 | 20.50 | 76.24 | 61.45 | 84.41 |

The major problem of these three tested algorithms is the final delineation of the individual PRA, which is the base for the coupling with dynamic avalanche models. The slope only and the Veitinger et al. (2016) algorithms do not delineate individual PRA. The Bühler et al. (2013) algorithm applies a basic delineation based on flow direction but the results are unsatisfactory because artificial straight lines of delineation occur (Figure 4). As dynamic avalanche models are very sensitive to PRA location and volume only a sophisticated delineation of the PRA can be applied for meaningful, scenario-based modelling.

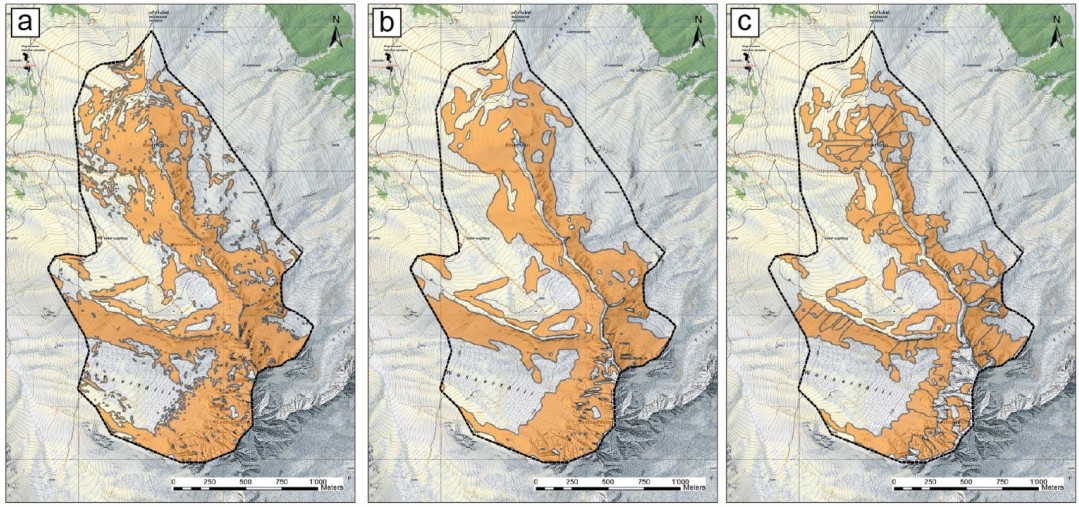

**Figure 4: results of the best parameter setting for the algorithms a) Voellmy (1955), b) Veitinger et al. (2016) and c) Bühler et al. (2013) for the test site Rinerhorn illustrating the missing or insufficient delineation of the individual PRA. (pixmaps©2018 swisstopo (5704 000 000), reproduced by permission of swisstopo (JA100118).**



### 3    Object Based Image Analysis approach

To overcome the issue with the PRA delineation, we develop a new algorithm based on object based image analysis (OBIA), originally developed to analyze remote sensing data (Blaschke, 2010) applying the Trimble eCognition Developer 9.3 software. The OBIA algorithm is based on the Bühler et al. (2013) algorithm and optimizes the delineation process.

**3.1 Input datasets**

To run the OBIA analysis we use the swissalti3D DEM provided by swisstopo, with a spatial resolution resampled from originally 2 m to 5 m. In several tests including manual evaluation with avalanche experts, we concluded that a 5 m resolution is sufficient to picture the terrain features relevant for avalanche release. We apply the elevation and its derivatives: slope angle, terrain ruggedness, aspect and fold (Figure 5). Additionally, we apply a forest layer, which is binary (forest/no forest).

The most important parameters and their settings are described here:

**Slope angle** (Burrough and McDonnell, 1998) is the first derivation of the elevation and describes the inclination of the terrain based on the elevation difference from the highest to the lowest adjacent cell in degrees and is a key parameter for avalanche release. To avoid a very unsettled picture caused by slope changes between single pixels, we filter the slope angle applying a

5 by 5 cell mean filter, weight by distance. This step eliminates isolated steep pixels as well as isolated flatter pixels. The filtered slope angle is a meaningful base to generate homogenous objects.

**Aspect** (Burrough and McDonnell, 1998) describes the downslope direction of the maximum rate of change from each cell to its neighbours. This describes the exposition of the terrain or the slope direction. We classify the DTM pixels into the eight

aspect sectors north, northeast, east, southeast, south, southwest, west and northwest. This is a key feature to delineate between the different PRA as a change in exposition usually limits the crack propagation (Van Herwijnen et al., 2016).

**Curvature** (Burrough and McDonnell, 1998) is the first derivation of the aspect or slope and parameterizes the convexity and concaveness of terrain. This can be done along the fall line of the slope (profile curvature) or across (plan curvature). We apply

the plan curvature to eliminate highly convex or concave areas limiting the fracture propagation of avalanche release (Van Herwijnen et al., 2016). The profile curvature is measured as the change in slope angle, the plan curvature as change in aspect.

**Ruggedness** (Sappington et al., 2007) is a slope angle independent measure for terrain roughness. Very rough terrain such as ridges and gullies prevent a widespread, connected weak layer and therefore the release of avalanches (Schweizer et al., 2003).

To calculate the ruggedness, a window size has to be specified. We apply a window size of 9 pixels, which corresponds to 45 m at an input resolution of 5 m. Within this window all normal vectors to the ground are analysed and their deviation is calculated. The result is a layer with normalized values between 0 (completely flat) and 1 (normal vectors diverge by 360°).



Rough terrain starts at values around 0.03 and after 0.08 the terrain is very rough. Values larger than 0.1 occur only very sparsely in natural terrain.

**Fold** describes the change of adjacent normal vectors and therefore is a good indicator for ridges and gullies as well as other

abrupt terrain changes (Schmudlach and Köhler, 2016). This is also key information to delineate between different PRA.

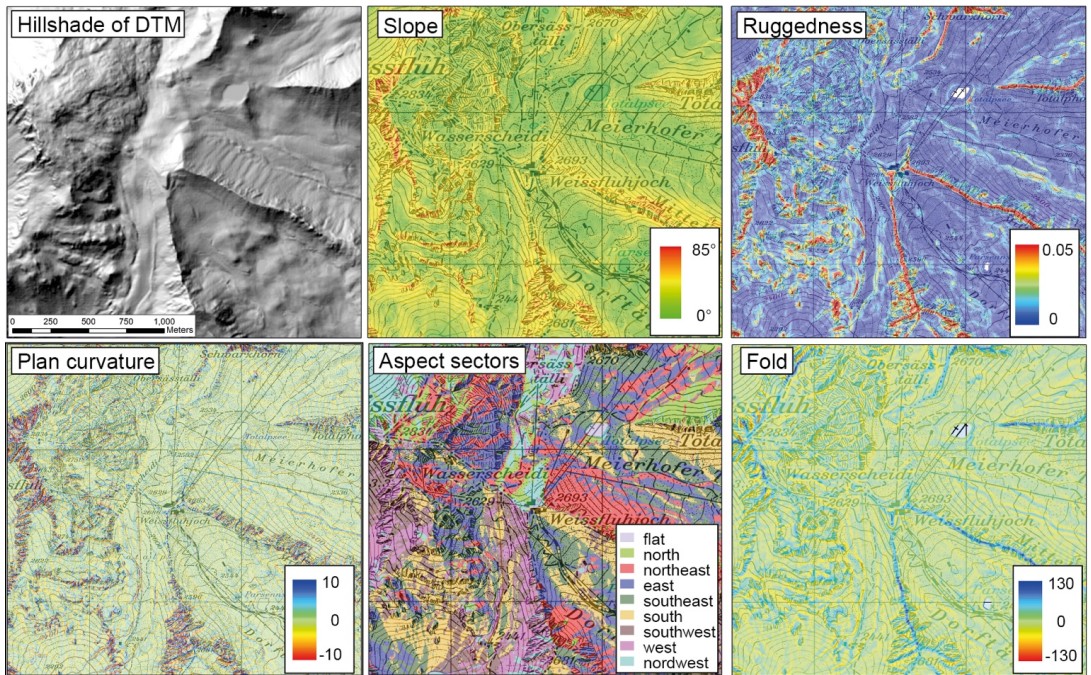

**Figure 5: Input data to delineate the PRA derived from the digital elevation model, at the Weissfluhjoch region in the ski resort Parsenn as an example (pixmaps©2018 swisstopo (5704 000 000), reproduced by permission of swisstopo (JA100118).**

**3.2  Processing steps**

To produce the PRA for the scenario *frequent*, we first identify steep slopes between 30° and 60° subtract then areas with high roughness (> 0.06) and high plan curvature values (> 6 rad/100 m) to eliminate gullies, ridges and rough rock faces, where no consecutive weak layer can develop and no large snow masses can deposit (Schweizer et al., 2003;Wirz et al., 2011). Additionally, isolated PRAs smaller than a certain area threshold are excluded. This first layer is then segmented and classified

into objects which are susceptible for PRA and NoPRA objects (Figure 6a).




In a second step, the susceptible area is further segmented with a finer scale parameter. We apply a multiresolution segmentation which takes into account variations in aspect sectors, slope and fold. We weigh variations in aspect sectors three times more than variations in slope and fold as changes in aspect sectors is the most important delineation parameters between individual PRA (Figure 6b). Finally, we classify PRA that are covered by forest (Figure 6c).

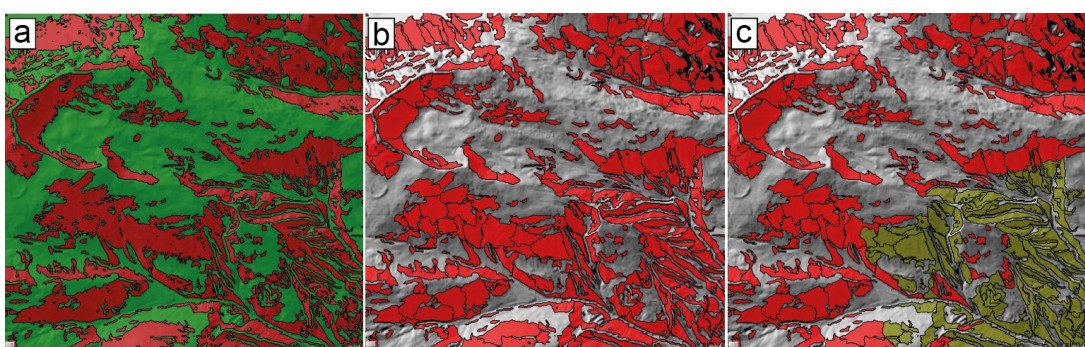

**Figure 6: Segmentation and classification of the terrain based on slope angle and ruggedness into PRA (red) and non PRA (light green) (a). Then the PRA are further segmented based on elevation, aspect, slope and fold (b). Finally, the PRA are classified as forest covered (dark green) and not forest covered (red) (c).**

To produce the PRA for the scenario *extreme*, we change the slope angle (28° - 60°) and the ruggedness threshold (> 0.08) to enable larger and more connected PRA (Figure 7a). Then we apply the same process tree as for the PRA in the scenario *frequent*. The small PRA are then classified after their median into the aspect sector classes (Figure 7b). Then we apply a region growing algorithm to merge adjacent objects with similar exposition, based on fold and slope angle. Finally, we classify

15 PRA that are covered by forest (Figure 7c).

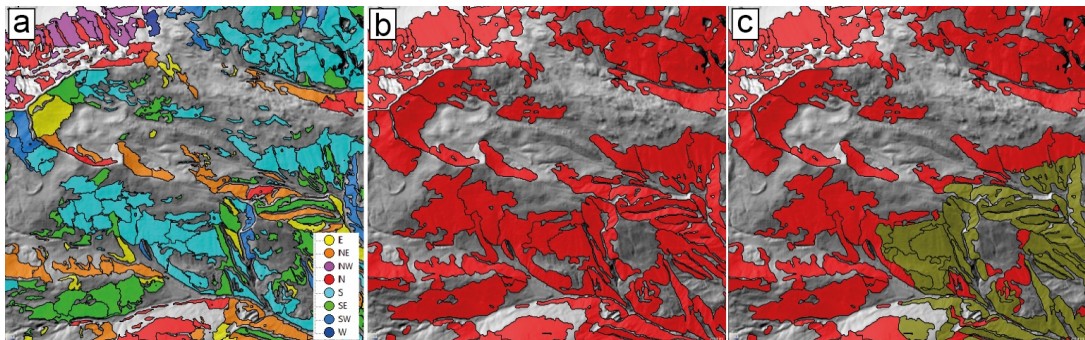





**Figure 7: Classification of the frequent PRA based on their most frequent aspect sectors (a). Region grow into object with similar exposition based on fold and slope angle (b). Classification of the forest covered objects in green (c).**

### 3.3 Results

To visualize the results of the OBIA based PRA algorithm we look at the greater region of Davos, with an extent of 20 by 25

5   km which equals 500 km². This area includes the three test sites Parsenn, Jakobshorn and Rinerhorn, depicted with dashed lines in Figure 8. Figure 9 gives a close up of the test site Rinerhorn. For the scenario *frequent* we get 16'167 individual release polygons. The mean area is 9'750 m² and the mean slope angle is 35.85°. For the scenario *extreme* we calculate 8'332 PRA with a mean area of 22'850 m² and a mean slope angle of 33.80°. Applying this approach, PRA for the scenarios *frequent* and *extreme* are directly connected. The *extreme* PRA consist of connected frequent PRA with similar exposition, fold and slope.

10  However, the total area is not exactly the same as we change the input layer thresholds for slope from 30° to 28° and the ruggedness threshold from 0.06 to 0.08 from the scenario *frequent* to the scenario *extreme*. This leads to larger and more connected PRA for the scenario *extreme*, which is in good agreement with the qualitative visual assessment performed by experienced SLF experts.

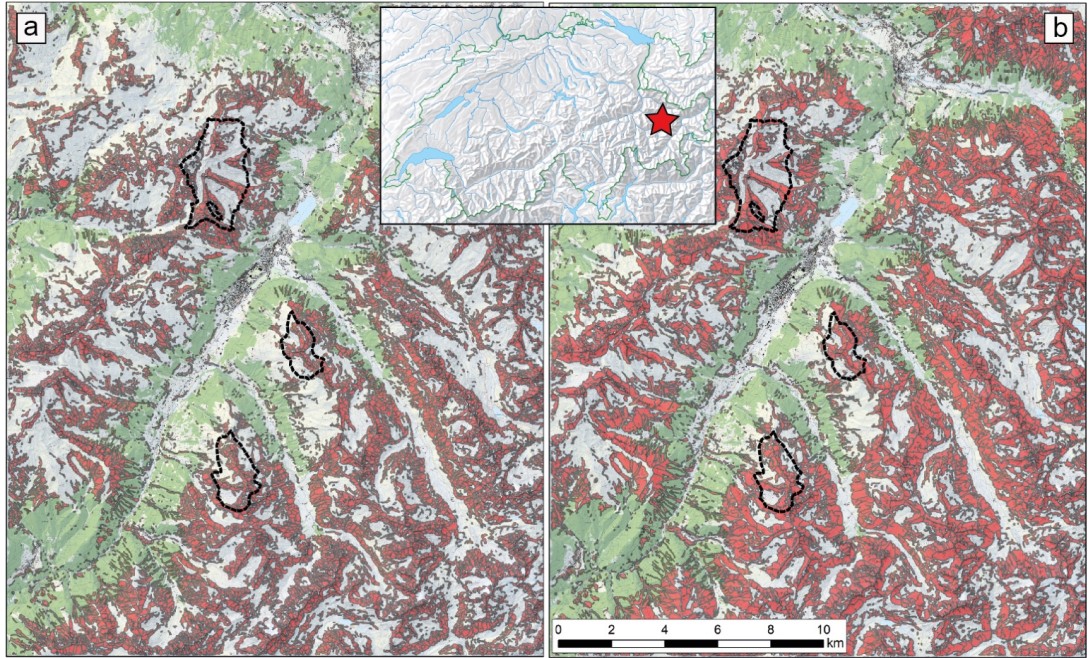

15  **Figure 8: Results for the OBIA based PRA algorithms for the scenario *frequent* (a) and the scenario *extreme* (b) without the PRA in forests. In dashed lines the extents of the test regions Parsenn (north), Jakobshorn (center) and Rinerhorn (south) are depicted (pixmaps©2018 swisstopo (5704 000 000), reproduced by permission of swisstopo (JA100118).**



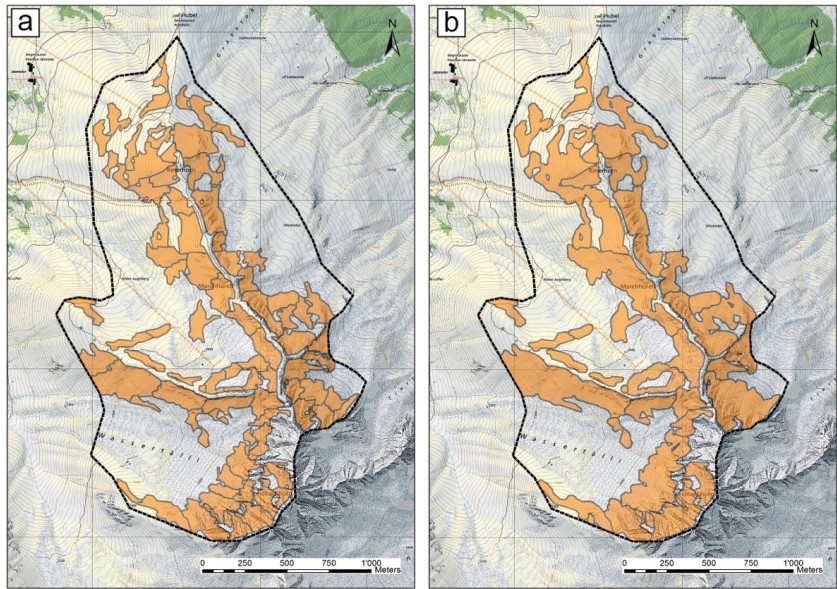

**Figure 9: Results of the OBIA algorithm for the scenario frequent (a) and the scenario extreme (b) at the test site Rinerhorn. (pixmaps©2018 swisstopo (5704 000 000), reproduced by permission of swisstopo (JA100118).**

The validation described in section 2.2 is now also applied to the OBIA algorithm (Table 3). The OBIA algorithm for the
5   frequent scenario is slightly better in POD and POFD than the Bühler et al. (2013) algorithm and exhibits an improved
delineation of the individual PRA. The comparison to the algorithms of Veitinger et al. (2016) and Voellmy (1955) to the
OBIA algorithm reveals a better performance which is quantified by the higher PSS and HSS scores and the lower amount of
total delineated area.

10   **Table 3: Validation results of all tested algorithms**

|  | POD [%] | POFD [%] | PSS [%] | HSS [%] | Area compared to slope 28-60° [%] |
|---|---|---|---|---|---|
| **Vollmey (1955), slope only** | 98.69 | 28.89 | 69.80 | 51.72 | 100.00 |
| **Bühler et al. (2013)** | 95.06 | 16.06 | 79.01 | 66.99 | 75.91 |
| **Veitinger et al. (2016)** | 96.74 | 20.50 | 76.24 | 61.45 | 84.41 |
| **OBIA frequent** | 95.22 | 15.57 | 79.65 | 67.85 | 75.07 |
| **OBIA extreme** | 97.84 | 19.78 | 78.06 | 63.21 | 83.72 |





### 3.4 Calculation of avalanche release depth (d0) and avalanche release volume

To perform an avalanche dynamics simulation not only the release area and its location is needed but also an average release depth, measured perpendicular to the slope. Combining these two pieces of information the avalanche volume can be calculated. In state-of-the-art avalanche dynamic models such as RAMMS, the applied friction values depend on the release

volume (Christen et al., 2010).

We implement the release depth calculation approach developed by Salm et al. (1990), which is applied for hazard mapping in Switzerland and is therefore well established. The estimation of release depth is based on the maximum snow depth increase within three days ΔHS(3), measured at automatic weather stations or study plots. These values are extrapolated using the

Gumbel extreme value statistics (Bocchiola et al., 2008). The longer the time series of snow depth measurements are and the more extreme weather events are captured, the more reliable the extrapolation gets. In Switzerland the maximum extrapolation time period applied is 300 years as it makes not much sense to further extrapolate applying measurement series not reaching back more than 85 years. However, in other countries, for example in Norway, this approach is applied to extrapolate as far as for 5000 years (Rudolf-Miklau et al., 2014). For the station Davos Flüelastrasse (1560 m a.s.l.), which we apply for this study,

the ΔHS(3) for the scenario *frequent* (10 year return period) is 0.82 m. For the scenario *extreme* (300 year return period) it is 1.44 m.

Now an elevation correction factor is applied to account for increasing snow depth with increasing elevation. In Switzerland ± 5 cm are applied per 100 m elevation difference. The work of Blanchet et al. (2009) found a value close to ± 2 cm per 100

m elevation difference, which might be more realistic for most regions in the Alps. To transform the ΔHS(3) measured at the flat field into the inclined slope of the release area, the values are corrected by multiplying it with the cosine of 28° in a first step. If necessary the snow depth value is increased by a factor for wind load. In practice values from 10 to 50 cm are used dependent on the expected additional wind load. Finally the correction for the inclination of the release area is applied for the final calculation of the fracture depth d0, using the function $\Psi = 0.219 / \sin(\alpha) - 0.202 * \cos(\alpha)$ of the slope angle $\alpha$. The basic

idea behind this approach is the observation, that steep slopes accumulate less snow than flatter slopes as the snow is on steep slopes less stable. This calculation steps are implemented in a Python script that calculates an individual release depth for every PRA polygon based on the mean elevation and the mean slope angle of the polygon. Based on the d0 value the release volume for every PRA is calculated for both scenarios. The mean d0 for the scenario *frequent* in the region Davos is 0.85 m and 1.60 m for the scenario *extreme*. The mean release volumes are 10'350 m³ (*frequent)* and 41'600 m³ (*extreme*).

### 3.5 Numerical avalanche dynamic simulations

The calculated release areas and the release depth define the avalanche release volume, necessary as input for the numerical avalanche dynamic simulations. We adapted the RAMMS::AVALANCHE software (Christen et al., 2010), applied for the





generation of hazard maps, to automatically process large number of release areas. This new module, RAMMS::LSHM applies the well-established friction parameter sets defined by Gruber and Margreth (2001) to all PRA polygons generated with the OBIA algorithm described in section 3. We split the PRA polygons into four volume classes as defined in Christen et al. (2010) and apply their specific friction parameters given in Table 4

5  **Table 4: RAMMS friction parameters applied for the simulation of the PRA for the scenarios *frequent* and *extreme***

| | Scenario *frequent* | | Scenario *extreme* | |
| --- | --- | --- | --- | --- |
| Volume categories | μ | ξ | μ | ξ |
| | (coulomb friction) | (turbulent friction) | (coulomb friction) | (turbulent friction) |
| **Tiny (< 5000 m³)** | 0.29 | 1500 | 0.275 | 1500 |
| **Small (5000 – 25'000 m³)** | 0.26 | 2000 | 0.235 | 2000 |
| **Medium (25'000 – 60'000 m³)** | 0.225 | 2500 | 0.195 | 2500 |
| **Large (> 60'000 m³)** | 0.18 | 3000 | 0.155 | 3000 |

The resulting maximum avalanche pressure values shown in (Figure 10) are later classified to a large scale hazard indication map.

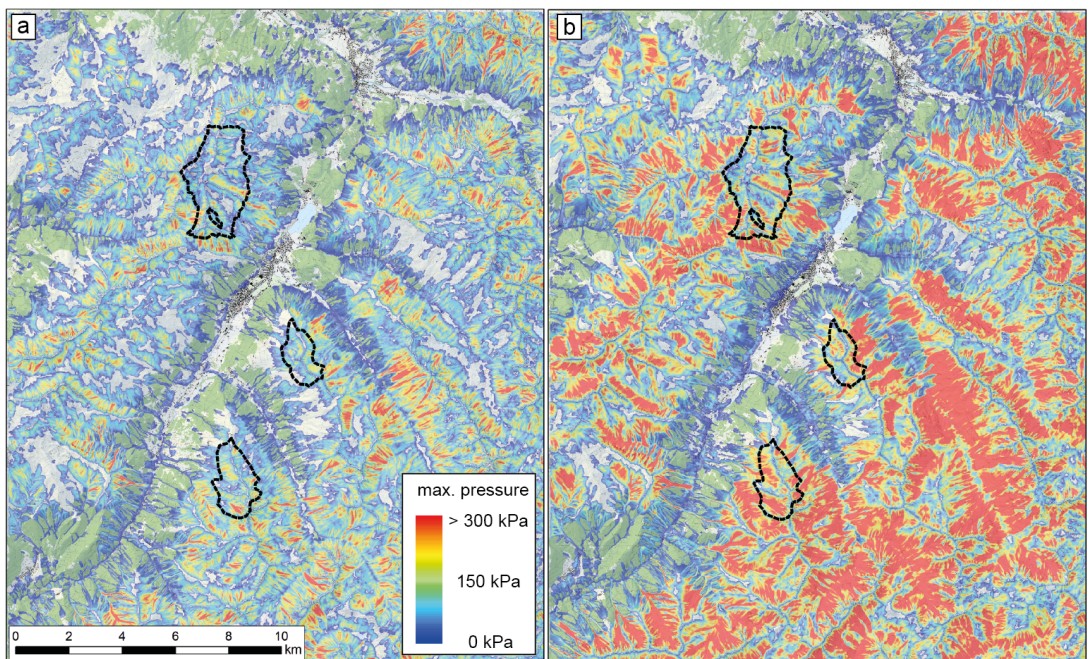

10  **Figure 10: RAMMS simulated maximum impact pressure values based on the PRA depicted in Figure 8 for the scenario *frequent* (a) and the scenario *extreme* (b). In dashed lines the extents of the test regions Parsenn, Jakobshorn and Rinerhorn are depicted (pixmaps©2018 swisstopo (5704 000 000), reproduced by permission of swisstopo (JA100118).**



## 4 Discussion

### 4.1 Validation of existing algorithms

The validation of PRA is a very difficult task as avalanche release areas often occur in poorly accessible terrain and may not be observed in time due to new snowfall or snow drift. Furthermore, accurate mapping of observed release areas is very

demanding in complex and steep terrain. In the region of Davos there are a lot of avalanche mapping activities performed by SLF and the local ski patrol staff but in most cases only the avalanche outline is mapped manually and not the release area specifically. The uncertainty concerning avalanches that have occurred but were not mapped is very high. Accurate avalanche mapping based on optical or radar aerial imagery or satellite data with sufficient spatial resolution can only be applied occasionally due to high data acquisition costs (Bühler et al., 2009;Lato et al., 2012;Eckerstorfer et al., 2016;Korzeniowska et

al., 2017). Therefore, no complete reference datasets over longer time periods are existing to our knowledge. To overcome this limitation and to enable a meaningful quantitative validation of the algorithms we produce a manually completed reference data set for three test sites (section 2.1). Since the automatic PRA delineation was carried out with regard to hazard indication mapping, high values for POD are requested to not miss PRAs that could produce destructive avalanches. For other purposes such as scenarios with very short return periods (1 – 5 years) usually applied for traffic line safety assessment, a lower POFD

is maybe more important. To enable a validation as objective as possible, we calculated the best parameter setting for every algorithm based on the reference data set (section 2.2).

The validation shows that the probability of detection (POD) is best with a value of 98.69 % for the simple slope angle approach even though not 100% are reached because of mapping errors in the reference data set and isolated PRA section above 60°.

However, the probability of false detection (POFD) is 28.89 % limiting its applicability for large scale hazard indication mapping because numerous unrealistic avalanches would be simulated. This leads to low overall accuracy measures (PSS = 69.80 %, HSS = 51.72 %).

The algorithm based on Veitinger et al. (2016) reduces the POFD to 20.50 % due to consideration of terrain roughness

(dependent on snow depth) but keeps a high POD of 96.74 %. These values indicate the suitability of this algorithm for large scale hazard indication mapping resulting in higher overall accuracy measures (PSS = 76.24 %, HSS = 61.45 %). This algorithm was developed to analyse avalanche release probabilities for smaller areas such as avalanche prone slopes next to single road sections in combination with snowpack and meteorological data for near real time hazard assessment and not for large scale hazard indication mapping. With the focus on single slopes, this algorithm does not delineate individual PRAs

which would be mandatory for the numerical avalanche simulations.

The algorithm based on Bühler et al. (2013) achieves a slightly lower POD of 95.06 % but an improved POFD of 16.06 %. Besides the roughness, this algorithm excludes also areas with high curvature values. This leads to the highest overall accuracy



measures of the three tested algorithms (PSS = 79.01 %, HSS = 66.99 %). Additionally, this algorithm tries to separate into individual PRA based on flow direction. But no scenarios can be depicted with this approach and the delineation of the individual PRA is unsatisfying.

The largest performance differences between the tested algorithms are not in the POD, which is very good for all algorithms, but in the POFD where we find considerable differences. Low POFD values mean that much less area is delineated as PRA, saving many time consuming numerical simulations. For example, the Bühler et al. (2013) approach results in 24 % less PRA area than the slope only approach.

### 4.2  Development and performance of the new object based image analysis (OBIA) approach

To overcome the limited possibilities, present in the tested algorithms for the delineation of the final PRA polygons, we develop a new PRA algorithm (section 3). With this algorithm, different scenarios with varying PRA sizes can be generated, which is a big advantage for large scale hazard indication mapping. The OBIA algorithm for the scenario *frequent* (5 – 30 years return period) achieves a POD of 95.39 % and the lowest POFD value of 15.88 %, eliminating most areas where avalanche do not occur. Therefore, this algorithm achieves the highest overall accuracy measures of all tested algorithms (PSS = 79.51 %, HSS

= 67.47 %). This result proofs the high performance of the newly developed OBIA algorithm for the *frequent* scenario.

Additionally, the OBIA algorithm was extended for an *extreme* avalanche scenario (100 – 300 years return period). The individual PRA grow into areas with minimum slope inclination of 28° and rougher terrain. They are generated in the same way as for the scenario *frequent*, but are then merged based on aspect, curvature and slope to obtain larger PRAs. This increases

the POD to 97.84 % but also increases the POFD to 19.78 %, the overall accuracy measures are reduced to PSS = 78.06% and HSS = 63.21%. Because the validation data sets contain mainly small to medium size PRA and only very few PRA of large avalanches, the validity of the reduced overall accuracy measures is limited. Unfortunately, no complete datasets for large PRA exist to our knowledge to meaningfully validate an extreme PRA scenario. Maybe this will change in the future as more cadaster information over longer periods get available and satellite imagery can be applied to accurately map extreme

avalanche cycles.

The OBIA algorithm is a novel and useful approach to generate two different PRA scenarios for large scale hazard indication mapping and enables regional to national scale applications. In particular in regions, where no or only limited avalanche cadasters exist and no experienced avalanche engineers have produced hazard maps, such an automated approach can be very

helpful for a preliminary hazard assessment. The delineation of the individual PRA is very difficult to validate. Compared to the algorithm of Bühler et al. (2013), only the OBIA algorithm performs a specific delineation of the individual PRA, and shows obvious improvements in particular within homogenous slopes (Figure 4 and Figure 9). This is achieved by the improved implementation of the aspect, curvature, fold and slope terrain characteristics into region grow algorithms within the




eCognition software. However, further investigations are needed to validate, refine and extend the delineation of the individual PRA, the definitions of the different scenarios and the adaption to specific local conditions.

### 4.3 Automated hazard indication mapping

Already in 2004 the project Silvaprotect performed automated avalanche dynamic simulations over the entire area of

Switzerland to identify protection forests (Gruber and Baltensweiler, 2004). At this time the digital elevation model available had only a resolution of 25 m (DHM25) and only a single scenario was calculated with a precursor version of RAMMS (AVAL-2D). The delineation of two PRA scenarios generated with the OBIA approach enables for the first time the calculation of numerical avalanche dynamic simulations over large areas with detailed terrain resolution. In combination with extrapolated extreme snow depth values describing potential release volumes, meaningful hazard intensity maps are generated that can be

easily translated into hazard indication maps. The procedure follows the simulation part that is applied for operational hazard mapping in Switzerland but can now be applied to areas up to several thousands of square kilometres also in regions where no hazard maps exist. A preliminary validation of the results with existing hazard maps in the canton Grisons, Switzerland (www.map.geo.gr.ch) and evaluation by local avalanche experts show a high quality of the automatically generated product (Figure 11). This approach opens the door for spatial continuous hazard indication information, which are today only available

in Switzerland with a very generalized level of detail. The direct comparison between the results of Silvaprotect (Gruber and Baltensweiler, 2004) and the approach presented in this investigation demonstrate that the results of the new approach are much more complete and accurate mainly due to the better DEM data and the refined PRA algorithm.

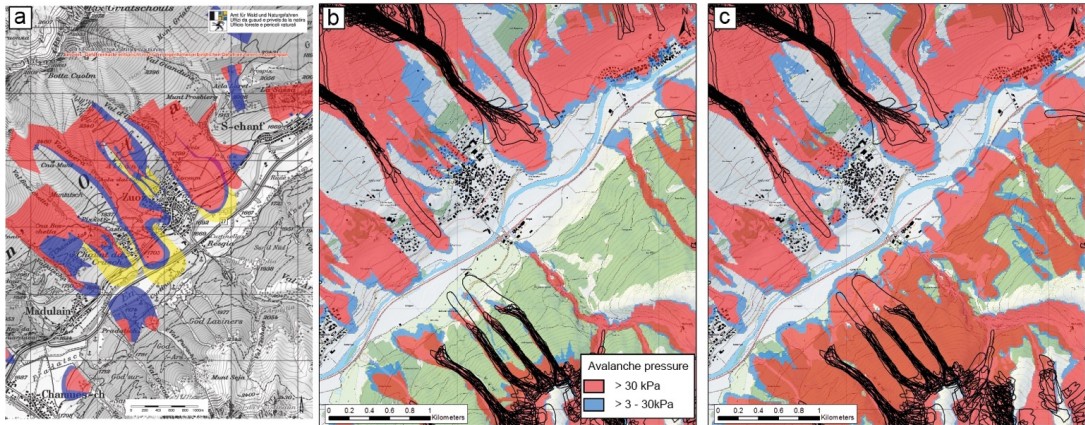

**Figure 11: Comparison of the official hazard map (a) with the results of the automated hazard indication mapping procedure with forest taken into account (b) and without forest taken into account (c) and the overlaid avalanche cadastre (black outlines) in the region of Zuoz, GR, Switzerland. (pixmaps©2018 swisstopo (5704 000 000), reproduced by permission of swisstopo (JA100118).**



The comparison of the automated hazard indication maps to the hazard maps and the cadastre information show in general a very good agreement (Figure 11). However, for some avalanche tracks such as the one in the middle of the map coming from the south, no hazard maps exist. This can be due to existing mitigation measures such as avalanche dams or snow supporting structures, which are not taken into account for the automated simulations or due to missing observations and therefore no

detailed hazard assessment was performed. But the major differences between the automated approach and the hazard maps can be explained by the applied forest layer. The protection forest is highly dynamic and the forest in high elevations in Switzerland shows a tendency to grow more dense (Bebi et al., 2009). Since the avalanche cadastre contains events recorded more than 60 years ago, changes in forest occurred at several locations. Most obvious is this for the two avalanche runouts from the cadastre reaching the valley floor in the south-western part of the map: The cadastre's runout distances are poorly

simulated in the scenario with forest but well modelled in the scenario without forest. These events have been observed in 1986 when the forest structure was less dense and contained more larch trees than today. This example illustrates the crucial role of forest information for large scale hazard indication mapping in regions with protection forests. In the future, with better up-to-date forest information derived from remote sensing (Waser et al., 2015), this source of error might get less important .

## 5 Conclusions and outlook

The development of automated PRA delineation algorithms based on digital elevation models (DEM) started in the early 2000s. As high quality DEM data gets more and more available even for mountain areas in remote regions (Bühler et al., 2012), such approaches have now the potential to be combined with numerical avalanche simulations to produce automated hazard indication maps. The validation of three different published approaches based on a nearly complete avalanche reference dataset from the region of Davos, Switzerland, reveals that the current detection performance of these algorithm is quite good

(PSS 69.80 – 79.01 %). The algorithms considering more than just slope angle improve the accuracy of the PRA delineation. Considering just slope angle works well for smooth terrain. For rough terrain, however, curvature and roughness provide essential additional information which should be considered in a successful algorithm. Important for the coupled numerical simulations is the total area delineated as PRA as every simulation consumes computational power and time. The tested algorithms reduce the total PRA by up to 24.09 % resulting in much less individual simulations and producing a more realistic

output. However, the delineation of the individual potential release areas (PRA) is insufficient and no connection to hazard scenarios is possible.

Therefore, we develop and validate a new PRA delineation algorithm, based on object-based image analysis (OBIA), which performs even better (PSS 79.51%), limits the total PRA area by 24.30 % and produces PRA with meaningful delineation. A

meaningful validation of the PRA delineation would be of great value. However, such reference data is to our knowledge not yet existing. With the OBIA approach it is possible to produce different hazard scenarios linked to return periods as the individual size of the PRA is variable. This is the prerequisite to produce meaningful hazard indication maps to automatically



evaluate avalanche hazard over large areas. The comparison to existing hazard maps shows a good agreement and illustrates the potential value of such maps in particular for regions where not much information and experience with avalanche hazard exists. In any case, up-to-date and accurate DEM data and information on the protection forest is crucial.

Our reference dataset is the most complete we know of considering PRA, however, it is only from a very limited region around Davos and does mostly contain smaller PRA. Therefore, we do not know how representative this dataset is for other regions. However, even though snow conditions may vary a lot between different locations, the basic terrain parameters leading to an avalanche release are estimated to be quite constant all over the world. Because we do not take information on the snow cover into account, we assume that our findings can be applied globally. But more research is necessary to proof this assumption.

Most important would be a validation for the *extreme* scenario with a dataset consisting of PRA from large avalanches. Unfortunately, such a dataset is not available today with sufficient quality.

On the long term, the current work could enable the coupling of terrain information, meteorological data, snow pack simulations and numerical avalanche simulations to achieve near real time hazard assessment over large areas as proposed by

(Vera Valero et al., 2016) and Veitinger and Sovilla (2016). However, the required input information in sufficient quality and resolution necessary for such a coupled system is very hard to get. In addition, the sensitivity of the individual information components has to be evaluated carefully. To obtain a reliable hazard assessment is therefore very difficult and we do not expect results that are applicable in practice in the near future. However, to achieve this goal on the long term, we encourage all research going into this direction.

**2   Acknowledgment**

The authors thank Roderick Kühne & Christian Willhelm from the Amt für Wald und Naturgefahren, Kanton Graubünden, for their technical and financial support. We thank the ski patrol heads of the Davos Klosters Mountains resorts for their support to improve the PRA reference datasets: Romano Pajarola (Parsenn), Vali Meier (Jakobshorn) and Nigg Conrad (Rinerhorn). We thank Betty Sovilla from SLF for the discussions on the parameter settings for the Veitinger et al. (2016) algorithm.

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
