# Peer review of "Automated snow avalanche release area delineation - validation of existing algorithms and proposition of a new object-based approach for large scale hazard indication mapping"

_Natural Hazards and Earth System Sciences, 2018_

## Referee Comment (RC1) · Anonymous Referee #1 · 1 Oct 2018

The manuscript "Automated snow avalanche release area delineation - validation of existing algorithms and proposition of a new object-based approach for large scale hazard indication mapping" by Y. Buehler and colleagues is dedicated to automated identification of snow avalanche hazard areas in a large scale.

Authors validated existing potential release algorithms (published in peer-reviewed journals), described the new approach based on object-based image analysis (OBIA) and prepared example numerical simulations in Davos region based on Swiss SLF RAMMS software.

The described topic is very important and actual for avalanche hazard specialists from all over the world. Especially from regions where there is not much information and experience in snow avalanche hazard.

The manuscript describes a very interesting, well tested and promising approach to cost-effective large-scale snow avalanche hazard mapping. All the new "OBIA procedure" steps are clear and based on the newest top-based international knowledge in this field.

A possible easy transfer of this approach to other regions (also outside the European Alps) is the main strength of this paper. From the scientific point of view, it seems to be a feasible strategy but it needs more research work (what is clearly mentioned by the Authors).

Given the complexity involved, the author has produced a number of positive and welcome outcomes including the literature review which offers a useful overview of current research and policy and the resulting bibliography which provides a very practical resource for current practitioners.

---

## Referee Comment (RC2) · Anonymous Referee #2 · 11 Oct 2018

**General comments:**

The research article 'Automated snow avalanche release area delineation – validation of existing algorithms and proposition of a new object-based approach for large scale hazard indication mapping' by Bühler et al. presents an improved approach to delineate hazard indication maps for larger areas. This is of high relevance to improve risk assessment but also to deepen our knowledge of on-site processes (when evaluating these hazard indication maps with at least some observations). It is always highly appreciated to improve large-scale 'modelling' in order to provide necessary fundamentals for vulnerable areas regarding a specific hazard - facilitating more targeted on-site assessments. The reasoning why an improved approach has to be developed is convincing and also the quality of the outcome. Of course, testing the approach for a broader range of influencing factors (topography, snow volume, and meteorological factors) would be necessary to verify the mentioned transferability to regions all around the world. Nevertheless, this would be a follow up for this article and I highly encourage the authors to keep track of spatially explicit modelling approaches and to steadily improve the presented approach.

Specific comments:

- I would appreciate a very short discussion whether this approach is only usable for snow slab avalanches or – to a certain degree – also for glide avalanches? Although the processes leading to the release of the avalanches are different, predisposition concerning (1) terrain, (2) snow volume, (3) meteorological factors are partly overlapping...probably. This aspect is shortly addressed on page 2, lines 1-4 but should be revisited in the discussion section.

- For the 'Input Parameter Setting' you have set one parameter to a 'Default Value' when changed systematically the other parameters. Why not changing all parameters for a specified range (using parameter sets instead of fixing numerous and changing one – as this could have interference effects). I assume that this was not done because of the enormous effort to implement parameter optimization routines (e.g. dream algorithm) and to link with the tested algorithms...but it should be addressed shortly in the discussion section if this could further improve your approach. Applying such types of adaptive Monte Carlo simulation could also lead to 'equifinality' of parameter sets and therefore this aspect, does not question the high quality of your approach and validation results but should be seen as another motivation to further work on the improvement of your approach.

- what were your criteria setting the default value? There might be some educated

guess, but please state (this comment is to a great extent linked to the comment above, asking myself if a different fixed default value would influence parameter optimization routine.

- Conclusions and Outlook section, line 25: put 'potential release areas (PRA)' in the first sentence and come up with PRA only at this point. It is nice to have the abbreviation PRA explained again this section, but at the very beginning.

СЗ

---

## Author Comment (AC1) · 1 Nov 2018

Dear Reviewer

Thank you for your positive comments.

We also presented our work at the International Snow Science Workshop ISSW 2018 in Innsbruck in October. We got many comments similar to yours from avalanche practitioners from around the world. By applying the methodology already in different regions in close collaboration with local avalanche experts, we gain more and more

experience on the strength and weaknesses of this methode by validation the results with avalanche observations.

---

## Author Comment (AC2) · 1 Nov 2018

Dear Reviewer

Thank you for your positive and encouraging review. Certainly, we keep on working on the spatial modelling. We apply the methodology in many different regions around the world in close collaboration with local avalanche experts. This testing and validation help us the further improve the algorithm and to stat to include more and more information on snow and weather conditions.

[Figure]

Specific reviewer comment 1: I would appreciate a very short discussion whether this approach is only usable for snow slab avalanches or – to a certain degree – also for glide avalanches? Al- though the processes leading to the release of the avalanches are different, predisposition concerning (1) terrain, (2) snow volume, (3) meteorological factors are partly overlapping...probably. This aspect is shortly addressed on page 2, lines 1-4 but should be revisited in the discussion section.

Answer 1: Thank you for this suggestion. We add the following sentence to the end of the discussion in the manuscript: In this research we present the processing chain for dry snow flowing avalanches. By incorporation information on snow temperature, snow erosion and free water content this approach could be extended with the scientific version of RAMMS (Bartelt et al., 2016;Bartelt and Buser, 2016) to simulate powder snow avalanches, wet snow avalanches, small skier triggered avalanches or glide snow avalanches. However, the validation of such simulations is very demanding in terms of valuable reference data but is planned for the future.

Specific reviewer comment 2: For the 'Input Parameter Setting' you have set one pa- rameter to a 'Default Value' when changed systematically the other parameters. Why not changing all parameters for a specified range (using parameter sets instead of fix- ing numerous and changing one – as this could have interference effects). I assume that this was not done because of the enormous effort to implement parameter opti- mization routines (e.g. dream algorithm) and to link with the tested algorithms: but it should be addressed shortly in the discussion section if this could further improve your approach. Applying such types of adaptive Monte Carlo simulation could also lead to 'equifinality' of parameter sets and therefore this aspect, does not question the high quality of your approach and validation results but should be seen as another moti- vation to further work on the improvement of your approach. What were your criteria setting the default value? There might be some educated guess, but please state (this comment is to a great extent linked to the comment above, asking myself if a different fixed default value would influence parameter optimization routine.

Answer 2: The value ranges were set as educated guesses based on previous publications (Schweizer, 2003;Bühler et al., 2013;Veitinger et al., 2016). Initially, the default values were set in the centre of the respective value range. Based on this, it was iterated over all parameters and subsequently the optimal parameter setting (highest skill score, explained in Bühler et al. (2018)) obtained. Afterwards, it was reiterated with the optimum values from the previous iteration as default values. If the optimal parameter setting was equal to the previous iteration and the corresponding skill score compared to the previous iterations the highest, the optimal values were considered as confirmed. In short: Starting with default values based on previous knowledge and taking the optimal values of the previous iteration as default values for the next iteration, we elaborated optimal values with the highest skill score.

Another approach would be, as you suggest, to evaluate every possible combination. However as the example of the algorithm of Bühler et al. (2018) shows, the evaluation of every possible combination would be very demanding in time:

Number of possible settings: Min slope angle: 20 Max slope angle: 20 Moving Window Roughness: 7 Max Roughness: 19 Plan curvature: 19 Minimal release Area: 10 Number of Combinations: 10'108'000

There are over 10 Mio possible combinations. The computational execution for one run and its evaluation takes about 2 minutes for test site. To evaluate all combinations, a computation time of about 1403 days would be needed.

Random parameter settings created with the help of Monte Carlo simulations could possibly be used to find new combinations with high skill score at random. The Application of DREAM-algorithms (Vrugt and Ter Braak, 2011) could further improve the calibration process. Running several chains with different starting points, the parameter value space could be explored efficiently and local optima (high skill score) could be found. However, this approach would be too time consuming for this study but could maybe be explored in a future study.

We add the following to the manuscript: The default values are set based on previously published investigations (Schweizer et al., 2003;Bühler et al., 2013;Veitinger et al., 2016).

Specific reviewer comment 3: Conclusions and Outlook section, line 25: put 'potential release areas (PRA)' in the first sentence and come up with PRA only at this point. It is nice to have the abbrevia- tion PRA explained again this section, but at the very beginning.

Answer 3: We change this in the manuscript as suggested.

References:

Bartelt, P., and Buser, O.: The relation between dilatancy, effective stress and dispersive pressure in granular avalanches, Acta Geotechnica, 11, 549-557, 10.1007/s11440-016-0463-7, 2016.

Bartelt, P., Buser, O., Valero, C. V., and Bühler, Y.: Configurational energy and the formation of mixed flowing/powder snow and ice avalanches, Annals of Glaciology, 57, 179 - 188, 10.3189/2016AoG71A464, 2016.

Bühler, Y., Kumar, S., Veitinger, J., Christen, M., Stoffel, A., and Snehmani: Automated identification of potential snow avalanche release areas based on digital elevation models, Natural Hazards and Earth System Science, 13, 1321-1335, 10.5194/nhess-13-1321-2013, 2013.

Bühler, Y., von Rickenbach, D., Stoffel, A., Margreth, S., Stoffel, L., and Christen, M.: Automated snow avalanche release area delineation – validation of existing algorithms and proposition of a new object-based approach for large scale hazard indication mapping, Nat. Hazards Earth Syst. Sci. Discuss., 2018, 1-24, 10.5194/nhess-2018-124, 2018.

Schweizer, J.: Snow avalanche formation, Reviews of Geophysics, 41, 10.1029/2002rg000123, 2003.

Veitinger, J., Purves, R. S., and Sovilla, B.: Potential slab avalanche release area identification from estimated winter terrain: a multi-scale, fuzzy logic approach, Natural Hazards and Earth System Sciences, 16, 2211-2225, 10.5194/nhess-16-2211-2016, 2016.

Vrugt, J. A., and Ter Braak, C. J. F.: DREAM(D): an adaptive Markov Chain Monte Carlo simulation algorithm to solve discrete, noncontinuous, and combinatorial posterior parameter estimation problems, Hydrology and Earth System Sciences, 15, 3701-3713, 10.5194/hess-15-3701-2011, 2011.